

# A multivariate statistical framework for mixed populations in compound flood analysis

Pravin Maduwantha[1,2], Thomas Wahl[1,2], Sara Santamaria-Aguilar[1,2], Robert Jane[1,2], James F. Booth[3], Hanbeen Kim[4,5], Gabriele Villarini[4,5]

[1]Department of Civil, Environmental and Construction Engineering, University of Central Florida, Orlando, FL 32816, USA
[2]National Center for Integrated Coastal Research, University of Central Florida, Orlando, FL 32816, USA
[3]Department of Earth and Atmospheric Sciences, City University of New York, City College, NY 10031, USA
[4]Department of Civil and Environmental Engineering, Princeton University, Princeton 08544, USA
[5]High Meadows Environmental Institute, Princeton University, Princeton 08544, USA

*Correspondence to*: Pravin Maduwantha (pravin@ucf.edu)

**Abstract.** In coastal regions, compound flooding can arise from a combination of different drivers such as storm surges, high tides, excess river discharge, and rainfall. Compound flood potential is often assessed by quantifying the dependence and joint probabilities of the flood drivers using multivariate models. However, most of these studies assume that all extreme

events originate from a single population. This assumption may not be valid for regions where flooding can arise from different generation processes, e.g., tropical cyclones (TCs) and extratropical cyclones (ETCs). Here we present a flexible copula-based statistical framework to assess compound flood potential from multiple flood drivers while explicitly accounting for different storm types. The proposed framework is applied to Gloucester City, New Jersey, and St. Petersburg, Florida as case studies. Our results highlight the importance of characterizing the contributions from TCs and non-TCs

separately to avoid potential underestimation of the compound flood potential. In both study regions, TCs modulate the tails of the joint distributions (events with higher return periods) while non-TC events have a strong effect on events with low to moderate joint return periods. We show that relying solely on TCs may be inadequate when estimating compound flood risk in coastal catchments that are also exposed to other storm types. We also assess the impact of non-classified storms that are neither linked to TCs or ETCs in the region (such as locally generated convective rainfall events and remotely forced storm

surges). The presented study utilizes historical data and analyzes two populations, but the framework is flexible and can be extended to account for additional storm types (e.g., storms with certain tracks or other characteristics) or can be used with model output data including hindcasts or future projections.

## 1 Introduction

Growing attention in scientific research has been directed towards compound extreme events resulting from various

hydrometeorological drivers, as their impacts are often more severe than those caused by univariate events (e.g., Wahl et al., 2015). Recent studies highlighted the threat of compound flooding in low-lying coastal and riverine regions that are



generally driven by the combination of precipitation, wind-generated storm surge, and streamflow (Hendry et al., 2019; Nasr et al., 2023; Wahl et al., 2015; Ward et al., 2018). Given the potentially devastating consequences of such events, the ability to quantify their likelihoods is crucial for flood risk assessments, infrastructure design, urban planning, (re-)insurance markets, and emergency response, among others.

There are two general methods that have been used in the literature to study compound flooding. The first one focuses on quantifying the compound flood potential by analyzing the dependencies and joint probabilities among compound flood drivers (Couasnon et al., 2020; Hendry et al., 2019; H. R. Moftakhari et al., 2017; Ward et al., 2018; Zheng et al., 2013). The second one focuses on quantifying compound flood hazard by employing physics-based models to obtain flood depths and spatial extents of historic flood events (Kumbier et al., 2018; Silva-Araya et al., 2018; Torres et al., 2015) or for large sets of synthetic events, where flood models are forced with boundary conditions of multiple flood sources (Bass & Badient 2018; Gori et al., 2020; Gori & Lin, 2022). Those boundary conditions can come from physics-based models (e.g., Gori et al., 2020) or statistical models (e.g., Jane et al. 2020). In either case, due to the requirement of numerous numerical model simulations, this approach is associated with large computational costs.

Calculating the probabilities of compound flood events is crucial and can be done either by applying extreme value analysis to flood depth information (Bass & Bedient, 2018; Gori et al., 2020; Gori & Lin, 2022) or by using multivariate models to quantify joint probabilities of the flooding drivers (e.g., Chen et al., 2012b, 2012a; Couasnon et al., 2020; Jane et al., 2020; Lian et al., 2013; H. R. Moftakhari et al., 2017; Sebastian et al., 2017; Zheng et al., 2014). For the latter, statistical techniques that have been used include Bayesian networks (Sebastian et al., 2017), Bivariate Threshold-Excess models (Zheng et al., 2014), Bivariate Point Process Method (Zheng et al., 2014), and Copulas (Chen et al., 2012a, 2012b; Couasnon et al., 2020; Jane et al., 2020; Lian et al., 2013; H. R. Moftakhari et al., 2017; Xu et al., 2018). Copulas have been extensively used to characterize the joint distribution of flood drivers due to their ability to model the dependence structure independently from the marginal distributions. However, most of these studies assume that all extreme events originate from a single population. This assumption may not be valid for regions where flood drivers can arise from different generation processes and mechanisms (e.g., (Smith et al., 2011; Kim et al., 2023). For example, both tropical cyclones (TCs) and extratropical cyclones (ETCs) can create extreme precipitation and extreme storm surges in the same coastal region, leading to compound flooding. Lai et al. (2021) estimated that TCs and ETCs are the major triggers of compound flooding. They found that more than 80% of compound flood potential in East Asia and more than 50% in the Gulf of Mexico are associated with TCs while ETCs contribute the most in Europe. TCs generally create more intense winds and rainfall (RF) compared to ETCs, while ETCs generally have greater spatial extents and can create RF over longer durations (e.g., Orton et al., 2016; Sinclair et al., 2020). Therefore, the flood drivers generated by these two storm types have different characteristics that may not be well captured by fitting them into a single probability distribution. Furthermore, Kim et al. (2023) highlighted that extreme events generated by TCs have a stronger correlation between RF and storm surge compared to the rest of the events they studied in the Dickinson Bayou watershed in Texas. This implies that the above-mentioned assumption of data coming from a single population could lead to a mischaracterization of the compound flood potential and/or compound flood hazard



(from hereon we use compound flood potential since the focus is on the statistical framework, but it can also be used as a starting point to assess compound flood hazard when coupled with a flood model).

Few studies addressed this aspect from the standpoint of coastal sea levels (i.e., univariate). For example, Orton et al. (2016) and Dullaart et al. (2021) quantified storm tide return periods by separately analyzing TCs and ETCs. In contrast, Lai et al. (2021) utilized copulas to model joint probabilities of flood drivers (RF and storm surge) for TCs and ETCs separately. Their study provides insights into the relative contribution of each storm type for joint probabilities but does not quantify the combined hazard. While Lai et al. (2021) and Kim et al. (2023) provided a starting point toward separating compound flood drivers by storm types, a comprehensive multivariate statistical framework for assessing compound flood potential from mixed populations does not currently exist.

An additional key aspect is analyzing compound flooding for future climate conditions where it will likely be amplified due to global warming (Bates et al., 2021). This is typically achieved by incorporating future sea level rise (SLR) projections and future storm climatologies derived from general circulation models (GCMs) (Bates et al., 2021; Bermúdez et al., 2021; Bevacqua et al., 2019; Gori & Lin, 2022; Khanam et al., 2021). However, the low resolution of available GCMs presents a challenge in capturing locally generated RF events, especially those related to convection (Imada & Kawase, 2021). Heavy precipitation can be caused by convection without being influenced by any cyclonic activity in the near atmosphere (Pfahl & Wernli, 2012). When combined even with small storm surges or just high astronomical tides, such events can lead to compound flooding where gravity-fed drainage is impeded by higher than normal coastal water levels. Therefore, it is important to understand the role of these types of events in driving compound flood potential, especially when including future projections from GCMs that are not capable of capturing them.

Prior studies employing synthetic TCs for compound flood analyses typically selected events that either crossed a specific search radius (e.g., Gori et al., 2020) or were generated in the study region (e.g., Bass & Badient 2018). However, cyclonic activity from distant systems can also generate moderate storm surges that propagate into the study region and, when combined with high tides and locally generated RF, can contribute to compound flooding. Hence, it is important to evaluate the potential mischaracterization of compound flood potential by neglecting such distant cyclonic events. Both of those questions regarding the role of local RF and remotely forced storm surge events have not been addressed in the literature.

This paper fills the above-mentioned gaps by introducing a copula-based statistical framework to estimate the compound flood potential at the catchment scale while accounting for mixed storm populations. We estimate the combined joint exceedance probabilities for different storm types and analyze the contribution of TCs, ETCs, and non-classified events (i.e., local RF and remotely forced storm surges). The framework is applied to Gloucester City, New Jersey, and St. Petersburg, Florida, as case studies.



## 2 Case study site and data

### 2.1 Study sites

Gloucester City is located in Camden County, New Jersey along the Delaware River (Fig. 1a) where it accommodates approximately 11,400 residents (Gloucester City New Jersey, n.d.). The city was affected by several major flood events in
the recent past generated by hurricanes and severe storms (Hurricane Floyd 1999, Hurricane Irene 2011, Hurricane Sandy 2012, and an unnamed storm in 2015). Its geographical location bounded by three rivers, the Delaware in the west, Newton Creek in the north, and Little Timber Creek in the south, makes the area susceptible to flooding from various sources. Our study area for Gloucester City encompasses two 14-digit hydrologic units (see Fig. 1b for the combined area) (Jones et al., 2022).

Located in Tampa Bay, Florida, and near the Gulf of Mexico (Fig. 1c), St. Petersburg is also exposed to coastal, fluvial, and pluvial flooding. The city was ranked among the top ten U.S. cities in terms of the highest asset value exposed to sea-level rise by 2070 (Nicholls et al., 2008). The economic average annual loss of St. Petersburg due to flooding was estimated as 244 million USD in 2005 and it is expected to increase to up to 763 million USD by 2050 under 20 cm sea level rise (Hallegatte et al., 2013). Our selected study area encompasses St. Petersburg and combines four 10-digit hydrologic units
(see Fig. 1d for the combined area) (Jones et al., 2022).

### 2.2 Data

The proposed methodology is based on historical data. We consider non-tidal residual (NTR) and RF as flood drivers. We use hourly water level data from the National Oceanic and Atmospheric Administration at the nearest tide gauge location of the study sites to obtain the NTR time series. For Gloucester City, we combine the data records of the Philadelphia (St. ID:
8545240) and Philadelphia Pier 11-north (St. ID: 8545530) tide gauges by adjusting for a 1 cm constant offset between records for the overlapping period to construct a 122-year-long data set from 1901 to 2021. The final record is nearly complete, with only 3% of missing data. The hourly water level data of the St. Petersburg tide gauge (St. ID: 8726520) from 1948 to 2021 was used for the St. Petersburg study area; missing data are less than 3%.

The water level time series are detrended using a 30-day moving average to remove the effects of relative mean sea level rise
and variability. Then, we perform a year-by-year harmonic tidal analysis using the Unified Tidal Analysis and Prediction (UTide) package in MATLAB to obtain the tidal constituents and tidal levels (Daniel Codiga, 2023). Years with more than 25% of missing data are omitted from the analysis (Philadelphia: 1903, 1921, 1922, and 1959; St. Petersburg: 1952 and 1964). Hourly time series of NTR are obtained by subtracting the predicted tidal levels from the observed water levels.

For RF, we use hourly gauge data from the longest records near each study site and combine it with gridded data from the
Analysis of Period of Record for Calibration (AORC) (Kitzmiller et al., 2018) in order to obtain spatial rainfall information and basin-averaged values (see Methods). The measured hourly RF gauge data records of Philadelphia International Airport and St. Petersburg start in 1900 and 1946, respectively. AORC RF data is constructed from different individual observed RF





data sets and available with an hourly temporal resolution and ~4 km spatial resolution covering the period from 1979 to the near present. It has been shown to have higher accuracy than other available gridded data sets when compared to in-situ

observations (e.g., Hong et al., 2022; Kim & Villarini, 2022).

We identify TC events using the HURDAT2 TC track data set from the National Hurricane Center to obtain the location of the center of circulation which is available in 6-hour intervals (Landsea & Franklin, 2013). To obtain the best tracks of ETCs, we use the MAP (Modeling, Analysis, and Prediction) Climatology of Midlatitude Storminess (MCMS) tracking algorithm (Bauer et al., 2016) on the ERA5 (fifth major global reanalysis produced by ECMWF data (Hersbach et al., 2020).

Considering the overlapping periods of available data sets (after combining gauge and AORC rainfall data), we perform the analysis for Gloucester City for the period of 1901 to 2021 and St. Petersburg from 1948 to 2021.

**Figure 1: Study site locations, selected catchment boundaries, locations of the rainfall gauges, tide gauges, and grid points of the AORC data for Gloucester City (a and b) and St. Petersburg (c and d).**



# 3 Methods

## 3.1 Bias correction of RF data

Following Kim et al. (2023), we use basin-averaged RF derived from all AORC grid points within the selected catchment areas. In addition, we want to leverage the long in-situ observations to obtain more robust results from the statistical analysis. Rain gauges measure very local weather conditions. However, the assumption that such point RF quantities are uniformly distributed over the entire catchment could lead to mischaracterization of the flood hazard potential. Therefore, we apply a bias correction to the hourly RF gauge data to match the hourly basin-average RF quantities calculated from AORC. The quantile mapping method is used for the bias correction, fitting both hourly measured gauge data and hourly AORC basin-averaged data to gamma distributions. We follow the approach outlined in Smitha et al. (2018) which can be mathematically expressed as:

$$RF_{Mod,x} = F_\gamma^{-1}(F_\gamma(RF_{MS,x}|\alpha_{MS}, \beta_{MS}), |\alpha_{AORC}, \beta_{AORC}) \tag{1}$$

where $RF_{Mod,x}$ is the bias-corrected measured value of the original value $RF_{MS,x}$, and $F_\gamma$ is the Gamma distribution with $\alpha$ and $\beta$ as scale and shape parameters. Fig. 2 shows the quantile plots before and after the bias correction for the two study sites. The bias-corrected hourly measured RF data are then aggregated to obtain RF accumulation time series ranging from 1 to 48 hours.

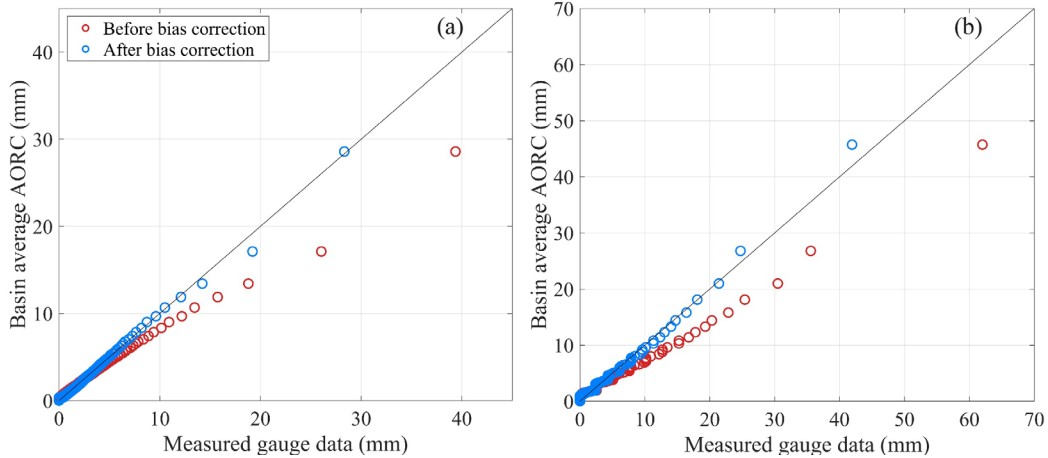

**Figure 2: Quantile plots between basin average AORC data and measured RF gauge data of (a) Gloucester City, and (b) St. Petersburg. The red circles show the quantiles before the bias correction and the blue circles after the bias correction.**

## 3.2 Extreme event sampling and stratification

We define extreme events that can potentially cause compound flooding using the peak-over-threshold (POT) approach. The threshold selection is a subjective process. The threshold needs to be high enough to lead to a good fit of marginal distributions and low enough to capture a sufficiently large number of events to obtain robust estimates of the distribution



parameters (i.e., bias-variance trade-off). Here, we set thresholds for NTR and RF time series such that we obtain 5 exceedances per year on average while using a 5-day declustering window (2.5 days before and after the event peaks) to ensure independence (Camus et al., 2021). We use the two-sided conditional sampling method outlined in Jane et al. (2020)

and adopted in Kim et al. (2023). When conditioned on NTR, the maximum RF value within a 3-day window is selected, and the same procedure is followed when conditioning on RF. The sampling process is applied for all the RF accumulation time series from 1 to 48 hours.

The identified POT events are stratified into two sets: events caused by TCs, and events not caused by TCs. An event is assumed to have been caused by a TC if the center of circulation of a TC passed through a 350 km search distance from the

center of the selected catchment within a time window of 3 days (2 days before and 1 day after) of a POT event. All other events are categorized as non-TC events. For Gloucester City, the threshold for NTR is set to 0.63 m resulting in a total number of 580 POT events (that is consistent with 5 events per year on average). For RF, thresholds are set to obtain 580 POT events for each RF accumulation time from 1 to 48 hours. After stratifying the POT events, 38 events are identified as TCs when conditioned on NTR and 43 when conditioned on RF, while the rest are non-TCs. For St. Petersburg, a threshold

of 0.34 m is used for NTR to obtain 355 POT events, and the same number of POT events is obtained for each RF accumulation time. When conditioning on NTR (RF), 37 (47) events are identified as TC-related.

To further sub-classify the non-TC events, we follow the same method using ETC track data but with a larger search radius of 1000 km in Gloucester City and 1200 km in St. Petersburg, reflecting the larger size of ETCs compared to TCs. The use of a 350-km radius for the TCs is likely to capture the large majority of TC-influenced events that generate extremes (Towey

et al., 2022).  The use of a 1000-km or 1200-km radius for ETCs is likely to capture nearly all cases where ETCs are involved (e.g., Towey et al., 2018). This gives three classes of events: TC, ETC, and non-classified events. When conditioned on RF, the non-classified sample mainly includes convective RF events that are not related to cyclonic activity in the near atmosphere. When conditioned on NTR, the non-classified sample includes, for example, NTR events that could have still been induced by TCs or ETCs which passed outside the search radiuses. In the subsequent analysis, we first focus

on the TC and non-TC events. Then we quantify changes in the joint probabilities when excluding non-classified events.

**3.3 Dependence analysis**

We calculate Kendall's $\tau$ between NTR and RF for all RF accumulation times from 1 to 48 hours to assess the sensitivity of the correlation to varying RF accumulation times (Kim et al., 2023). This is done in three ways: (a) all the POT events without stratification, (b) TC events, and (c) non-TC events. We find the RF accumulation time corresponding to the

maximum correlation and use it for the bivariate analysis. To be consistent with the annual exceedance probability estimation process, a single RF accumulation time is selected in all stratified samples of a given study site.

Some regions along the U.S. coast experienced an increase in the correlation between RF and NTR since the mid-20th century (Wahl et al, 2015). Therefore, to assess temporal changes in the dependence, Kendall's $\tau$ is calculated for 30-year moving windows, shifted 1-year each time step (see Fig. 4). We assess the significance of temporal changes using the range



of natural variability. The range is calculated as the 5th and 95th percentiles of Kendall's $\tau$ values obtained from randomly sampling 30 years of data for 10,000 iterations (Wahl et al., 2015). When a calculated $\tau$ value falls outside this range, the change is considered significant. The non-stationarity analysis is conducted using the selected RF accumulation time for each case study location.

## 3.4 Marginal distributions and joint probability analysis

Next, we identify the best-fitting marginal distributions for each set of stratified POT samples (TC and non-TC). The conditioning variables (both NTR and RF) are fit to the generalized Pareto distribution (GPD), which is most suitable to model POT extremes. When conditioning on NTR, the corresponding maximum RF sample is fit to various distributions with a lower bound at zero, and the best model is selected using the Akaike information criterion (AIC) (Akaike, 1974). Here, the Birnbaum-Saunders, exponential, two-parameter gamma, three-parameter gamma, inverse Gaussian, lognormal,

Tweedie, Weibull, two-parameter mixed gamma, and three-parameter mixed gamma distributions are tested. When conditioned on RF, the corresponding maximum NTR sample is fit to logistic and Gaussian distributions.

We use copulas to model the joint dependence between NTR and RF. According to Sklar's theorem (Sklar, 1959), the bivariate cumulative distribution $F_{XY}(x, y)$ of the variables $X$ and $Y$, with univariate marginal distributions $F_X(x)$ and $F_Y(y)$, for all $(x, y) \in \mathbb{R}^2$, can be written as

$$F_{XY}(x, y) = C[F_X(x), F_Y(y)] \tag{2}$$

where function C represents the bivariate copula on [0,1]. Annual exceedance probabilities (AEP) can refer to different hazard scenarios (e.g., AND, OR, Survival Kendall, Structural) that define different geometries of Upper Sets that contain the events perceived as "dangerous" (Salvadori et al., 2016). Considering the recommendations of Moftakhari et al. (2019) for compound flood assessments, we use the "AND" scenario which represents the exceedance of both $X$ and $Y$. The joint

AEP of a given pair of $(x, y)$ is calculated as

$$AEP_{(x,y)} = P(X > x \cap Y > y)/\lambda = (1 - F_X(x) - F_Y(y) + C_{XY}(x,y))/\lambda \tag{3}$$

$$C_{XY}(x, y) = C[F_X(x), F_Y(y)] \tag{4}$$

where $\lambda$ is the average inter-arrival time between threshold exceedances. We consider 40 possible copula families plus the independent copula using the *VineCopula* R package (Nagler et al., 2023) to identify the best-fitting copula family for each

pair of samples. The most appropriate copulas are selected based on AIC.

After fitting the selected copulas to the stratified two-sided POT samples, the joint AEP of a given pair of $(NTR, RF)$ can be calculated as follows:

$$AEP_{(NTR,RF)}^{TC,con.NTR} = (1 - F_{(NTR)}^{TC,con.NTR} - F_{(RF)}^{TC,con.NTR} + C_{(NTR,RF)}^{TC,con.NTR})/\lambda^{TC,con.NTR} \tag{5}$$

$$AEP_{(NTR,RF)}^{TC,con.RF} = (1 - F_{(NTR)}^{TC,con.RF} - F_{(RF)}^{TC,con.RF} + C_{(NTR,RF)}^{TC,con.RF})/\lambda^{TC,con.RF} \tag{6}$$

$$AEP_{(NTR,RF)}^{non-TC,con.NTR} = (1 - F_{(NTR)}^{non-TC,con.NTR} - F_{(RF)}^{non-TC,con.NTR} + C_{(NTR,RF)}^{non-TC,con.NTR})/\lambda^{non-TC,con.NTR} \tag{7}$$

$$AEP_{(NTR,RF)}^{non-TC,con.RF} = (1 - F_{(NTR)}^{non-TC,con.RF} - F_{(RF)}^{non-TC,con.RF} + C_{(NTR,RF)}^{non-TC,con.RF})/\lambda^{non-TC,con.RF} \tag{8}$$





where $F$ is the marginal distribution and $C$ is the copula function. Following Bender et al. (2016) (and many other studies since then), we derive the combined AEP of a selected population (TC or non-TC) for a given pair of $(NTR, RF)$ by taking the maximum AEP from the two conditioned samples.

$$AEP^{TC}_{(NTR,RF)} = max\left\{AEP^{TC,con.NTR}_{(NTR,RF)}, AEP^{TC,con.RF}_{(NTR,RF)}\right\} \tag{9}$$

$$AEP^{non-TC}_{(NTR,RF)} = max\left\{AEP^{non-TC,con.NTR}_{(NTR,RF)}, AEP^{non-TC,con.RF}_{(NTR,RF)}\right\} \tag{10}$$

### 3.5 Combining joint exceedance probabilities of two populations

The calculated AEPs from Eqs. (9) and (10) provide the joint AEPs for NTR and RF associated with the two populations, TC and non-TC, separately. However, both TC and non-TC events can create compound flooding in the same catchment. In the stratification process, a given POT event was either categorized as caused by TC or non-TC thus making the probability distributions of these two populations independent from each other. Accordingly, the total annual non-exceedance probability ($ANEP$) of a given pair of $(NTR, RF)$ can be calculated as follows:

$$ANEP_{(NTR,RF)} = ANEP^{TC}_{(NTR,RF)} \times ANEP^{non-TC}_{(NTR,RF)} \tag{11}$$

$$ANEP_{(NTR,RF)} = \left(1 - AEP^{TC}_{(NTR,RF)}\right) \times (1 - AEP^{non-TC}_{(NTR,RF)}) \tag{12}$$

The associated return period ($RP$) is calculated as

$$RP_{(NTR,RF)} = \frac{1}{1 - ANEP_{(NTR,RF)}} \tag{13}$$

To perform the above calculations for generating joint probability isolines, the parametric space is discretized into small intervals along both NTR ($NTR_1, NTR_2, ..NTR_i, ...NTR_n$) and RF ($RF_1, RF_2, ..RF_j, ...RF_m$) axes, creating a mesh. AEPs on each point on this mesh are calculated for all pairs of $(NTR_i, RF_j)$.

To quantify the relative probabilities of events along certain quantile isolines, we obtain 10,000 combinations of NTR and RF by sampling from the fitted copulas such that the relative proportion of extremes is consistent with the empirical distribution. Then the relative probability along the isolines is calculated by the kernel density function of the simulated sample. The location of the "most likely" event is assigned to the point with the highest relative probability density on an isoline (Salvadori & Michele, 2013).

### 3.6 Role of non-classified events

For assessing the contribution of events that are non-classified, we repeat the same steps outlined above stratifying the POT samples in four ways: (1) both conditional samples are stratified as TCs and non-TCs, (2) sample conditioned on RF is stratified as TC and ETC, and sample conditioned on NTR is stratified as TC and non-TC; (3) sample conditioned on RF is stratified as TC and non-TC and sample conditioned on NTR is stratified as TC and ETC; and (4) both conditional samples are stratified as TC and ETC. More specifically, in (2) we omit the non-classified POT RF events (77 in Gloucester City and 192 in St. Petersburg), assuming that those are mainly convective events that are not captured, for example, by coarse GCMs





and hence missing from compound flood assessments for future climates. In (3) we omit the non-classified POT NTR events (43 in Gloucester City and 105 in St. Petersburg), which could be caused by distant storms outside our search radius or, in the case of Gloucester City, could be influenced by high discharge in the Delaware River. In (4) we omit both types of non-
classified events mentioned in (2) and (3).

To assess the impact of omitting different types of non-classified events in (2), (3), and (4), we keep the calculated joint probability distribution of (1) as a reference and calculate the relative change in AEP along the probability isolines of (1). We also keep the copula family from (1) fixed for (2), (3), and (4) to isolate effects unrelated to switching to a different copula; results for the same analysis but allowing for different copula families to be selected are shown in Supplementary
Material. We perform this analysis for the period from 1950 to 2021 where ETC tracks are available and both ETC and TC track data are more reliable.

## 4 Results

### 4.1 Dependence analysis

We use Kendall's rank correlation coefficient $\tau$ to calculate the strength of dependence between NTR and RF for different
RF accumulation times. For both study sites, NTR and RF exhibit a weak but statistically significant correlation when considering all the POT events without any stratification (Fig. 3). However, when the events are caused by TCs the correlation is stronger and varies with the RF accumulation time. Considering both conditioned samples, 18-hr and 16-hr RF accumulation times are selected for the bivariate statistical modeling in Gloucester City and St. Petersburg, respectively.

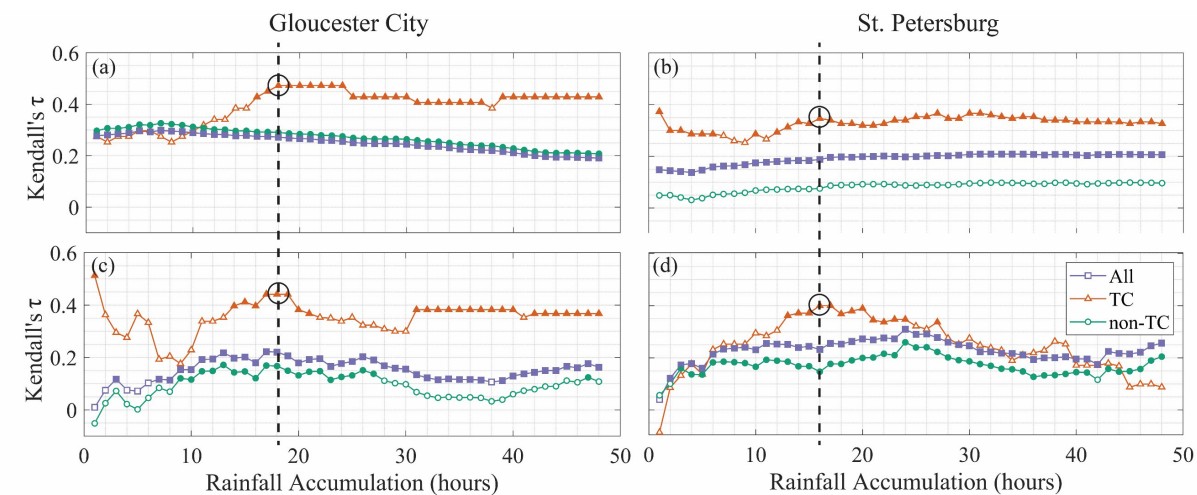

**Figure 3: Kendall's τ between NTR and RF for different RF accumulation times for all events (purple), TCs (orange), and non-TCs (green) for samples conditioned on NTR (a and b) and RF (c and d). Filled markers indicate values that are significant at 5% level. The black circles with vertical dashed line show the selected RF accumulation for each location.**



When testing for non-stationarity in the dependence we focus on the selected RF accumulation times. Fig. 4a shows a
significant change in τ during the last three decades at Gloucester City. A similar change is not detected for St. Petersburg,
but the correlation values are more frequently significant during the late 20th and early 21st century (Fig. 4b). Therefore, to
reflect current climate conditions and avoid underestimation of compounding effects, we use only the last 30 years of data to
model the dependence structure at both study sites.

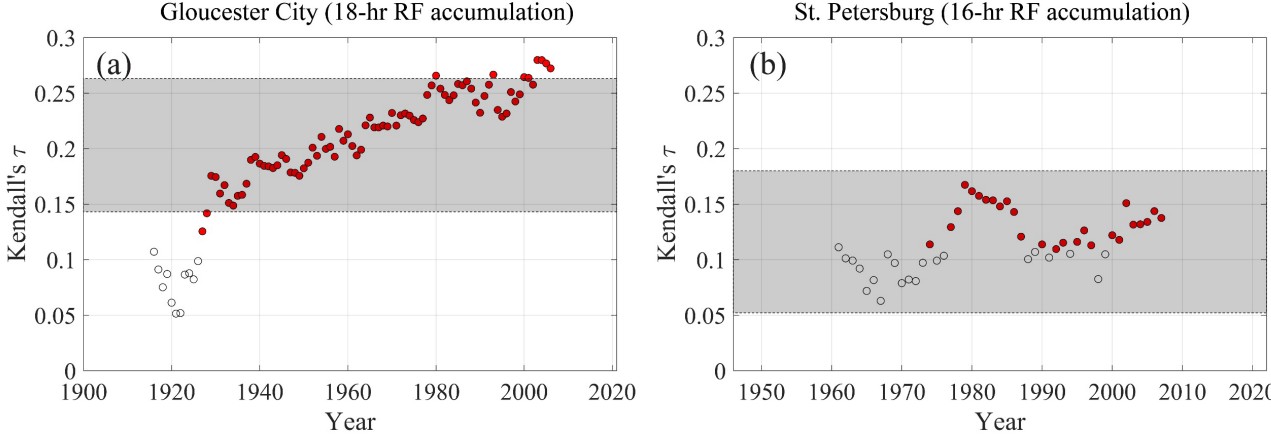


**Figure 4: Changes in Kendall's τ between NTR and RF derived from a 30-year moving time window for (a)
Gloucester City (18-hr accumulation) and (b) St. Petersburg (16-hr accumulation). Each circle represents the
midpoint of the 30-year window. Red circles indicate significant correlation (α = 0.05). Grey horizontal bands
represent the range between the 5th and 95th percentiles of natural variability.**


## 4.2 Bivariate statistical analysis

As described in Section 3.4, the conditioning variable is fit by a GPD and for the conditioned variable several parametric
distributions are tested. Selected distributions for each sample are indicated in Fig. 5 and Fig. 6 for the two study sites. The
confidence interval of the empirical cumulative distribution function (CDF) is calculated using the Dvoretzky–Kiefer–
Wolfowitz inequality (Dvoretzky et al., 1956).




**Figure 5: Selected parametric distributions compared to the empirical distributions of Gloucester City for TC events (left) and non-TC events (right). (a), (b) NTR POT events when conditioned on NTR. (c), (d) Maximum RF events corresponding to NTR POT events when conditioned on NTR. (e), (f) Maximum NTR events corresponding to RF POT events when conditioned on RF; (g), (h) RF POT events when conditioned on RF. The red dots represent the empirical Cumulative Distribution Function (CDF) of the observations. Dashed lines denote 95% confidence intervals of the empirical CDF calculated using the Dvoretzky–Kiefer–Wolfowitz inequality.**





**Figure 6: Same as Figure 5, but for St. Petersburg.**

The quantile isolines of 5, 10, 20, 50 and 100-year return periods are obtained for each conditional sample for the two study

locations (Fig. 7 and Fig. 8). The number of events in the stratified samples and the selected copula models are indicated in

each panel. A relatively lower number of TC events are captured in both conditional samples compared to non-TC events

 

(Fig. 7 and Fig. 8). The return periods of events with extreme NTR and non-extreme RF are lower (the AEP is higher) when they are caused by TCs compared to non-TCs. When the samples are conditioned on RF, events with extreme RF and non-extreme NTR show nearly the same return periods in both populations (panels (b) and (e) of Fig. 7 and Fig. 8).


**Figure 7: Results of bivariate statistical analysis for Gloucester City. Numbers in the top-right corner of panels (a), (b), (d), and (e) indicate the number of POT events in each sample and the selected copula family. Quantile isolines of the 5, 10, 20, 50, and 100-year joint return periods are shown where the color scale indicates the relative probability of events along the isolines. Panels (c) and (f) show the combined results for both conditioned samples. Note the**
**different x- and y-axis scales for better clarity.**





**Figure 8: Same as Figure 7 but for St. Petersburg, FL.**

The quantile isolines after combining the joint probability distributions of the two storm type populations (TC and non-TC) are shown in Fig. 9. To quantify the relative contribution from each of them, we calculate the ratio of AEP contributed by TCs to the total AEP along the isolines (Fig. 10). In Gloucester City, more than 60% of the AEP of low probability events (i.e., high return periods, e.g., RP = 50 or 100) are associated with TC events while more than 70% of the AEP of high probability events (low return periods, e.g., RP = 5 or 10) are associated with non-TCs. In St. Petersburg, when both NTR

and RF are extreme, TCs mainly drive the joint AEP. For example, TCs contribute over 90% to the AEP of events with more than 100 mm 16-hr RF and 0.8 m NTR. Closer to the axes (either extreme NTR or extreme RF) both types of events contribute approximately evenly to the joint AEP for rare events. For both locations, the region with extreme RF and non-extreme NTR shows a major contribution of non-TCs to the AEP.



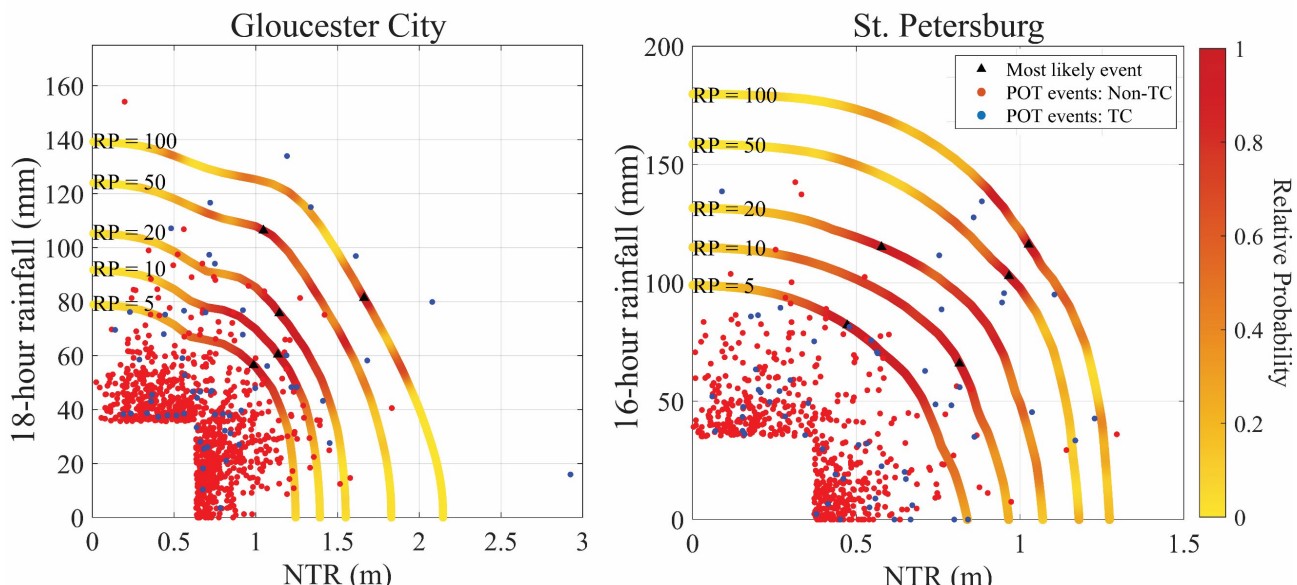

**Figure 9: Joint probability isolines for the two study sites after combining the AEPs of the two populations (TC and non-TC). The color scale indicates the relative probability of events along the isolines. The location of the "most likely" event is assigned to the point with the highest relative probability density on an isoline.**

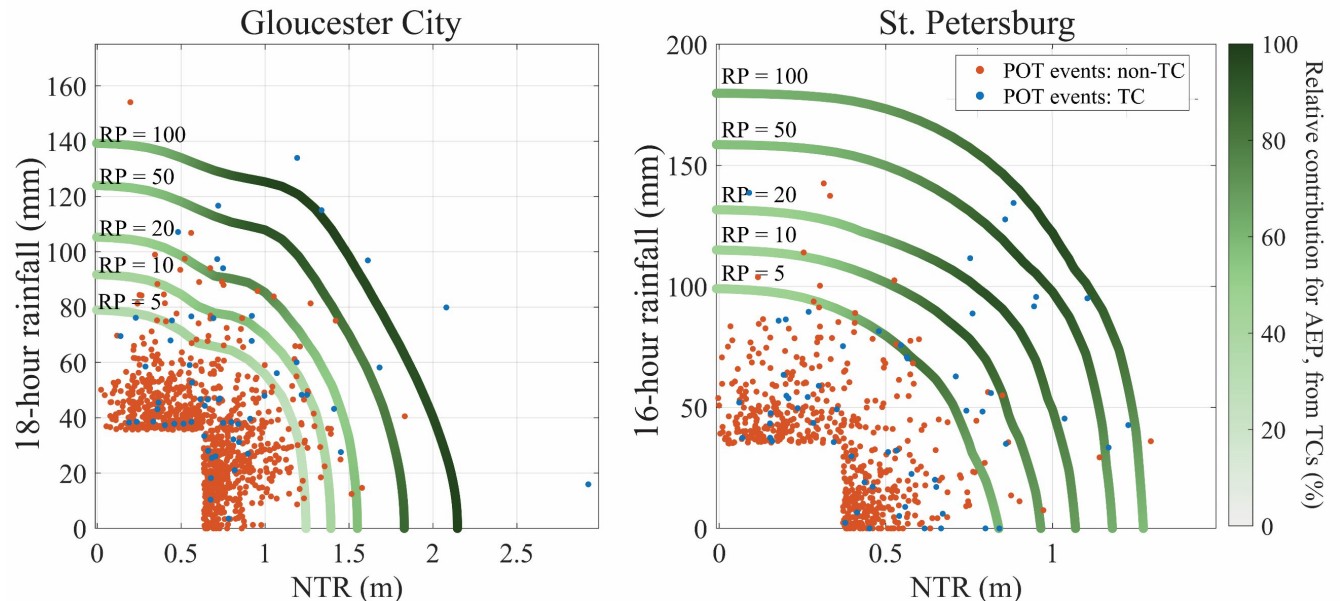

**Figure 10: Relative contribution of TCs to the AEP along the joint probability isolines for the two study locations. The color scale indicates the ratio of AEP associated with TCs to the combined AEP of both populations.**





### 4.3 Role of non-classified events

To assess the role of non-classified events, such as convective RF events and surge events caused by remote cyclones, we
repeat the above analysis as outlined in Section 3.6. The calculated relative changes in AEP along the joint probability
isolines after combining the AEPs of the two populations are shown in Fig. 11. When removing non-classified POT RF
events (those are mainly convective events), the AEP decreases for events where RF is extreme and NTR is small to
moderate (Fig. 11a, b). This becomes more noticeable for events with higher AEPs (lower return periods, e.g., RP = 5 or 10),
especially in St. Petersburg where it reaches up to 25% reduction. However, in Gloucester City, this impact becomes
negligible for rare events (RP = 50 or 100). When the non-classified POT NTR events (e.g., caused by remote cyclones) are
removed from the analysis, the AEP slightly decreases for events where NTR is extreme and RF is small, while the AEP
slightly increases when NTR is extreme and RF is moderate (Fig. 11c, d).

As mentioned in Section 3.6, we assume that the copula types that are used for the previous analysis (with events stratified as
TC and non-TC) remain the same after removing non-classified events. We also conducted the analysis while allowing the
selected copulas for the different conditional samples to change (see Fig. S1 and Table S1). When allowing copulas to
change, the changes in the AEPs along the isolines become more pronounced in Gloucester City when removing the non-
classified POT NTR events. For example, they increase up to 35% for the events where NTR is extreme and RF is moderate
(Fig. S1 e).









**Figure 11: Change in AEP of combined populations in Gloucester City (left) and St. Petersburg (right): (a), (b) when omitting the non-classified POT RF events; (c), (d) when omitting the non-classified POT NTR events; and (e), (f) when omitting all non-classified events. The change in AEP was calculated along the joint probability isolines derived for the analysis where events are stratified as TC and non-TC.**





# 5 Discussion

## 5.1 Dependence analysis

The strong correlation between NTR and RF when the events are caused by TCs (see Fig. 3) suggests that there is a higher potential for compound flooding by TCs in both study sites. This can be attributed to the nature of TCs, notably their propensity of extreme RF potential combined with strong winds. The non-TC samples in our analysis contain all events that are not directly linked to a TC (e.g., ETCs or convective events) and show weak but stable (over different rainfall accumulation times) correlation between NTR and RF. The stronger TC correlation is in line, for example, with findings reported by Kim et al. (2023) for the Dickinson Bayou watershed in Texas. However, they reported weak and insignificant correlation throughout all RF accumulation times (1 to 48hrs) for non-TC events when conditioning on NTR. For Gloucester City the same conditional sample shows slightly higher and significant correlation for all RF accumulation times (see Fig. 3). When the events are not stratified and treated as a single population, the correlation is similar to the one found for non-TCs leading to an underestimation of the compound flood potential. These results highlight the importance of differentiating between storm types associated with different physical processes and to characterize their individual contributions to compound flood potential. By selecting an RF accumulation time that leads to the maximum correlation we account for the sensitivity of correlation to RF accumulation time while following a conservative approach that avoids underestimating the dependence and compound flood potential.

## 5.2 Combining joint exceedance probabilities of two populations

Prior studies have only focused on merging the AEPs of two populations within a univariate framework (e.g., Orton et al., 2020). Our proposed methodology combines the AEPs of two populations in a bivariate framework that also has the flexibility to be further expanded to account for three or more populations. The joint probability distributions (Fig. 9), as determined by combining the AEPs of two populations, provide insights into the compound flood potential at each of the study sites. The framework derives a single isoline for each return period which can be used, for example, to derive a single "most likely" design event (Salvadori et al., 2011, 2013). For Gloucester City, for instance, the 100-yr most-likely design event (i.e., the event with the highest relative probability along the 100-yr isoline) is given by 1.66 m NTR and 81 mm 18-hr RF. Under the assumption of independence between NTR and RF the same event has a return period of 173 years. It should be noted that the Philadelphia tide gauge is located along the Delaware River (see Fig. 1) and the recorded water levels are influenced by both wind-driven storm surge and river discharge. In this study, we do not explicitly account for wind-driven surge and river discharge separately (NTR incorporates both), but the framework presented here is flexible enough to be extended for this purpose, e.g., by including NTR from an open coast tide gauge near the Delaware River mouth and an upstream stream gauge. The most-likely 100-yr design event for St. Petersburg is comprised of 1.03 m NTR and 116 mm 16-hr RF; under the independence assumption the return period of the same event increases to 155 years.



The magnitude of the univariate 100-yr 18-hr RF event is (after combining the AEPs of two populations) 139 mm for Gloucester City and 180 mm for St. Petersburg (see Fig. 9). This is approximately an 11% increase compared to the univariate 100-yr 18-hr RF of the TC samples at each site (125 mm and 163 mm). For the univariate 5-yr 18-hr RF the increase is 49% in Gloucester City (from 53 mm to 79 mm) and 34% in St. Petersburg (from 78 mm to 99 mm) (Fig. 9).

The return levels for RF are similar for the TC and non-TC samples but TCs lead to higher NTR return levels, particularly at Gloucester City (see Fig. 10). This implies that the most extreme surges are caused by TCs but both TC and non-TC events can produce similar rainfall totals. These results indicate that relying solely on TCs may be inadequate when analyzing compound flood risk in coastal catchments that are also exposed to other storm types. TCs are most relevant for events with low AEPs, which are important for flood zoning and critical infrastructure design. ETCs, on the other hand, mainly drive

compound flood potential for events with higher AEPs, which are most relevant, for example, for storm water management and design.

### 5.3 Role of non-classified events

As described in the Section 3.6, the analysis was repeated to assess the impact of non-classified events. This including, for example, NTR events caused by remote storms outside our pre-defined search radiuses, NTR in Gloucester City that is

caused by high river discharge, and convective RF events. This is important because when studying future compound flooding, we typically rely on GCMs, which do not capture convective RF events. Similarly, remotely triggered NTR events may be missed in assessments where only cyclones that pass through a defined region are included. Our analysis sheds light on the relative importance of those types of events in addition to TCs and ETCs that directly affect the area of interest.

Neglecting RF events that are likely locally generated can lead to an underestimation of the overall AEP (or overestimation

of the return period), particularly for more frequent events (RP =5, 10) (see panels (a) and (b) of Fig. 11). This can be attributed to the nature of locally generated convective RF events since they are generally less intense in magnitude but occur with a greater frequency compared with TCs and ETCs. The reduction in AEP is also higher in St. Petersburg where 55% (192 events) of the RF events exceeding the threshold are not linked to ETCs or TCs. This highlights the importance of convective RF events in St. Petersburg and thus the need for high resolution models to characterize future flood hazard

potential.

Removing non-classified events where NTR exceeded the threshold from the analysis leads to a slight reduction of AEP for events where NTR is extreme and RF is small, and to a slight increase in AEP where NTR is extreme and RF is moderate (see panels (c) and (d) of Fig. 11). This increase in AEP is potentially caused by the fact that the non-classified events have weaker correlation between NTR and RF compared to ETCs and TCs, hence excluding them leads to larger joint AEPs. In

the specific case of Gloucester City, those non-classified events can also include high discharge events in the Delaware River leading to high water levels (and hence high NTR) at the Philadelphia tide gauge.

Overall, excluding non-classified events leads to smaller changes in the joint probabilities in Gloucester City compared to St. Petersburg, especially when focusing on rare events (RP = 50,100) (see panels (e) and (f) of Fig. 11). This is likely because



of the proximity of Gloucester City to the U.S. East Coast where ETCs are more frequent and the overall number of non-
classified events is lower (77 when conditioned on RF, 43 when conditioned on NTR).

**6 Conclusions**

This paper introduces a flexible copula-based statistical framework to assess compound flood potential from multiple drivers
while explicitly accounting for different storm types. Here we apply the method to two case study sites, Gloucester City,
New Jersey, and St. Petersburg, Florida, which are exposed to different storm climatology. Our study highlights the
importance of differentiating between storm types with different physical processes and characterizing their individual
contributions to compound flood potential. Overall, we find that TCs modulate the tails of the joint distributions (events with
higher return periods), while non-TC events have a strong effect on events with low to moderate joint return periods. While
the most extreme events and associated probabilities are most relevant for flood zoning and critical infrastructure design,
more moderate events are crucial, for example, for stormwater management and design. We also quantify how non-classified
storms that are neither linked to TCs or ETCs in the region impact the compound flood potential. The results differ across
study sites with the effects being overall smaller in Gloucester City than in St. Petersburg, and they also affect different parts
of the joint distribution differently. This is important because assessments of future compound flood hazard (and flood
hazard in general) typically rely on the output from GCMs that may not capture such events due to their coarse resolution.
This can in turn lead to a misrepresentation of flood hazard and risk. Our results provide insights into how large (or small)
the effect is of not capturing all relevant storm types when studying compound flooding. The method is flexible so that
additional storm types (e.g., storms with certain tracks or other characteristics) can be identified and their effect on
compound flood potential can be quantified. Finally, while we focus on observed data and its derivative products, the
framework can be used with model output data including hindcasts or future projections.

The focus here is on the compound flood potential and how the joint probabilities of different flood drivers are linked to
different storm types. However, the combined probability distributions of the different populations can also be used to
generate a large number of synthetic events that can act as boundary conditions for hydrodynamic numerical models to fully
characterize compound flood hazard, including flood depths and extend. This will be demonstrated in a separate study.

**Code availability**

The marginal distribution fitting and copula selection was done using the MultiHazard R package which can be downloaded
from GitHub at https://github.com/rjaneUCF/MultiHazard (DOI: https://doi.org/10.5194/nhess-20-2681-2020.). The other
codes are available in GitHub at https://github.com/CoRE-Lab-UCF/MACH-Compound-Flooding. (The DOI and the final
version of the codes will be available after addressing the reviewers' comments and suggestions.)



**Data availability**

The measured rainfall data used in this paper can be downloaded through NOAA's National Climatic Data Center's (NCDC) archive of global historical weather and climate data at https://www.ncdc.noaa.gov/cdo-web. The AORC (4-km) Version 1.1 datasets can be obtained from the NOAA computer system and are available at https://hydrology.nws.noaa.gov/aorc-historic/. The hourly water level data at Philadelphia (St. ID: 8545240, St. ID: 8545530) and St. Petersburg (St. ID: 8726520) can be accessed through National Oceanic and Atmospheric Administration (NOAA: http://tidesandcurrents.noaa.gov/). The
HURDAT2 data are available from https://www.nhc.noaa.gov/data/hurdat. ETC track data set is available upon request from the authors.

**Author contribution**

The study was conceived by TW and PM. PM developed the methodology, undertook the analysis, and wrote the first draft of the paper under the guidance of TW, SSA and RJ. JFB provided the stratified ETC data. HK and GV contributed to
technical discussions in the early stages of the analysis. All authors co-wrote the final manuscript.

**Competing interests**

Authors declare that they have no competing interests.

**Acknowledgment**

PM, TW, and SSA were supported by the National Science Foundation as part of the Megalopolitan Coastal Transformation
Hub (MACH) under NSF award ICER-2103754. JFB was supported by NSF award 1854773. RJ and TW acknowledge financial support from the USACE Climate Preparedness and Resilience Community of Practice

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
