# Peer review of "A multivariate statistical framework for mixed storm types in compound flood analysis"

_EGUsphere, 2024_

## Referee Comment (RC2)

[referee-annotated manuscript omitted]

---

## Referee Comment (RC3)

I would like to compliment the authors for a well-written paper. The research gap this paper aims to fill is clear and relevant. Here are some minor comments to consider

1) Consider adjusting the title to speak to a larger audience. Currently, the term 'mixed populations' does not speak for itself. Instead, it would be good to focus on the fact that you look at different storm types that can drive compound flooding
   a. As a follow-up, the term 'populations' is never clearly defined.
2) As RC1 and RC2 have highlighted there are some limitations and uncertainties in the methods that you have used. In the paper, I am missing a proper discussion of these limitations and how to overcome them in the discussion/conclusion

---

## Author Comment (AC1)

**Reviewer 1:**

This paper discusses making use of copulas and extreme value distributions conditional on different storm types to quantify annual exceedance probabilities of joint events at multiple locations, with a focus on two case studies in the USA. Events are described as being driven by non-tidal residuals or by rainfall caused by, in part, tropical and extratropical cyclones. The paper looks to analyse the relationships between flooding impact and these different drivers based on meteorological observations (ERA5) and hydrological modelling (UTide). Extreme events were identified using a Peaks-over-Threshold approach with a simple Independence/declustering criterion, and attributed to cyclones with a simple spatio-temporal proximity metric. Joint AEPs are based on "AND" scenarios (occurrence of events at all locations at the same time) and use a variety of copula families to determine these.

Overall, looking at the two similar problems of event driver attribution and simultaneous events is fairly novel. Although the constituent parts are known standards, the combination is not one this reviewer has come across before, which is much to the paper's benefit. At times, this paper is spinning a lot of plates at once, but ultimately the conclusions are clearly shown by the methods and data. The stated flexibility to more than two sites simultaneously is mentioned but not demonstrated.

Overall, this is a strong paper, which needs some adjustments to improve the communication of the methods and the conclusions.

The authors would like to thank the Reviewer for providing thoughtful comments. Below, the Reviewer can find our responses to each comment, including how we will address each of them in the revised manuscript.

Major issues:

The POT event declustering is quite simplistic and doesn't count for the varying speed of response of the flow to rainfall at each location.

Thank you for the comment. We agree that the varying speed of the runoff affects flood depths in the catchment. However, the objective of this paper is to assess the compound flood potential, in other words the probability of different combinations of rainfall and coastal water levels to co-occur. The physical response of the catchments to these compound rainfall and NTR events is therefore out of the scope of this study, and it will need to be assessed using numerical flood models that are able to represent many characteristics of the catchments that can affect resulting flooding (such as elevation, slope, soil type for infiltration and friction). The output of the framework we propose can act as boundary conditions for these numerical flood models.

The reference to "high" and "low" return periods from line 330 onwards needs to be more specific. The exact choice of cutoff is not important, but stating what it is and sticking to it is important.

The sentences will be changed as follows specifying the respective range of return periods within the brackets.

The sentence starting from L331:

"To quantify the relative contribution from each of them, we calculate the ratio of AEP contributed by TCs to the total AEP along the isolines (Fig. 10). In Gloucester City, more than 60% of the AEP of low probability events (i.e., events with return periods above 50 years) are associated with TC events while more than 70% of the AEP of high probability events (i.e., events with return periods below 20 years) are associated with non-TCs. In St. Petersburg, when both NTR and RF are extreme, TCs mainly drive the joint AEP. "

The sentence starting from L353:

"This becomes more noticeable for events with higher AEPs (i.e., events with return periods below 10 years), especially in St. Petersburg where it reaches up to 25% reduction. However, in Gloucester City, this impact becomes negligible for rare events (i.e., events with return periods above 50 years)."

The sentence starting from L434:

"Neglecting RF events that are likely locally generated can lead to an underestimation of the overall AEP (or overestimation of the return period), particularly for more frequent events (i.e., events with return periods below 10 years) (see panels (a) and (b) of Fig. 11)."

The sentence starting from L447:

"Overall, excluding non-classified events leads to smaller changes in the joint probabilities in Gloucester City compared to St. Petersburg, especially when focusing on rare events (i.e., events with return periods above 50 years) (see panels (e) and (f) of Fig. 11)."

Using high numbers of different copulas and distributions can make it hard to compare results, especially between sites. Using more general distributions (e.g. kappa) or more general copulas and sticking to one overall best choice may make it easier to interpret.

Our ultimate aim is the robust estimation of return levels which we believe is best achieved by selecting the best copula for each location with its particular dependence structure. As the results show, the correlation between the flood drivers can vary considerably across sites and between the TC and non-TC samples. Therefore, imposing a single copula family (which may not be a good fit in all cases) could mischaracterize the dependence structure. While it may make cross-site comparisons more straightforward, we would rather obtain the best results possible for each location. Then, when comparing the results of two sites, the derived probability distributions at each site are used, which are derived using the four best copulas for each stratified-conditional sample.

Minor issues:

The heavy use of abbreviations does make this harder to read, consider sometimes switching back to fully spelling out terms like rainfall.

We agree with the Reviewer that the overuse of abbreviations might make reading difficult. Therefore, we have removed those abbreviations that were used less than 25 times along the manuscript (e.g., Akaike information criterion (AIC) and general circulation models (GCMs)). All others are used extensively.

Check your references in the main body match those in the bibliography.

Thank you. The bibliography was re-checked and corrected.

Some paragraphs could be swapped for tables (paragraph around line 168, paragraph around line 330

We will revise the section at line 330 to define low/high return periods (RPs) as suggested by the Reviewer, and now we believe it is clearer in the paragraph format. Additionally, there is now more information that may not well fit into a tabular format.

For the section around line 168, we propose including a table that compares metrics for Gloucester City and St. Petersburg. This table would facilitate a direct comparison by listing the thresholds and number of peak-over threshold events. However, this addition may increase the length of the manuscript, as the table would need to be positioned above the corresponding text. Therefore, we kept the paragraph as in the original manuscript and hope this meets your approval.

line 252: numbered lists should be presented as such, with a new line for each for readability.

Agreed, the text has been changed as recommended by the Reviewer.

Figure 4: It is unclear what is exactly meant by "natural variability", and how it differs from the significance shown by the red circles.

In the manuscript, Figure 4 shows the changes in Kendall's $\tau$ between non-tidal residuals and rainfall derived from a 30-year moving time window. Each circle represents the calculated Kendall's $\tau$ for the peak-over threshold events during the respective 30-year period. The circles are marked in red when Kendall's $\tau$ is significant ($\alpha = 0.05$) for the respective 30-year period. We define the natural variability by randomly sampling 30 years many times (10,000 iterations were used) and calculating Kendall's $\tau$. The range (gray band) corresponds to the 95$^{th}$ and 5$^{th}$ percentiles of the calculated Kendall's $\tau$ from the sampling process and it provides information about how $\tau$ could change by random chance over time due to natural variability. This is different from the individual $\tau$ values being significant or not.

We have added a sentence (in L279) in section 4.1 to clarify how the natural variability is defined and calculated.

"Fig. 4 shows the changes in Kendall's $\tau$ between NTR and RF derived from a 30-year moving time window and the range of the natural variability of the correlation. We define the natural variability by randomly sampling 30 years many times (10,000 iterations were used) and calculating Kendall's $\tau$ for each sample."

Figures 7 and 8 are very busy. Consider splitting into more figures or remove elements of the figures which do not directly contribute to your conclusions.

Thank you for the comment. For more clarity, the layout is changed to three rows and two columns (previously it was 2 by 3). The number of events in each panel and the name of the selected copula family were removed and will be added to the supplementary material (table).

---

## Author Comment (AC2)

**Reviewer 2:**

This paper is well-written and scientifically sound. I would like to see more discussion on the following topics:

The authors would like to thank the Reviewer for their insightful comments. Below are our responses to each comment, including the planned adjustments to the manuscript as per the NHESS review process.

[1] you end up with results for RF and NTR. Please recommend how to combine these results in a flood risk analysis

The main objective of the paper is to provide a framework that can be used to assess the compound flood potential and to derive joint probabilities of flood drivers accounting for the different statistical characteristics depending on the storm type that generates them. Procedures for combining the return level estimates among different populations in a univariate analysis are available (Barth et al., 2019), however in the bivariate space it is an active area of research with most similar studies focusing on a single population of events (e.g., TCs in Kim et al., 2023).

Flood risks can be estimated using different approaches/definitions. If we assume the IPCC definition, flood risk is a combination of flood hazard, with exposure and vulnerability. Our results can be used to estimate the flood hazard. Again, there are several approaches that can be followed. As a common approach, combinations of RF and NTR of specific probabilities (e.g. 100 years) can be selected to derive the resulting flooding from these events using a flood model and thus estimating the flood hazard. In another approach, the statistical analysis is performed on some response variable (e.g., flood depth at a given location) after running a large number of events comprised of synthetic RF and NTR through a flood model. The authors are currently working on a framework to produce time series of NTR and rainfall fields for these synthetic events, as they are required as boundary conditions by numerical flood models. To summarize the above process, the following part is added to the discussion section (where the limitations are discussed).

"To extend the proposed framework for fully characterizing the compound flood risk, the statistical approach can be combined with hydrodynamic numerical models (so called hybrid modeling; e.g., Moftakhari et al. 2019) to estimate flood inundation. However, analyzing only the most likely event (even though it may be the most plausible given the observations) does not capture the range of flood levels that could be generated by different combinations of flood drivers (NTR and RF) along an isoline. One way to address this limitation is to sample an ensemble of events (peak NTR-RF combinations) along the isoline and run them through flood models. Alternatively, a response-based approach can be employed, which involves simulating flood hazard for a large number of synthetic events from the multivariate statical model and then performing the statistical analysis on the response variable of interest (e.g., flood depth at a given location). The latter is computationally demanding, possibly necessitating the use of a surrogate model, however the return level estimates are likely to be more robust than when adopting an event-based approach (Jane et al., 2022). The simulated probabilistic flood depths and extents can then be incorporated with exposure and vulnerability data to perform a comprehensive flood risk assessment."

[2] the isolines method has some shortcomings in my view. The paper hints that these can be used to define a single event (the one on the isoline with the highest probability density), However, in most

compound flood risk analysis there is no single "most representative event". In coastal zone there are locations close to the coastline for which the main flood driver is the peak sea level (surge) and there are locations more inland for which the rainfall is the dominant flood river. For the former, an event with extreme NTR and moderate RF is the most relevant, for the latter an event with extreme RF and moderate NTR is the most relevant.

Thank you for the very useful comment. In this paper we adopt the "most-likely event" strategy, introduced by Salvadori et al. (2011) and subsequently used in other studies (e.g., Jane et al., 2020), as a straightforward method to derive possible design events associated with a given return period. Analyzing just the most likely event (even though it may be the most plausible given the observations) will not capture the different flood levels that could be generated through the different combinations of flood drivers (NTR and RF) along an isoline. One option to overcome this limitation is to stick with the event-based approach and to sample an ensemble of events along the isoline, and then run them through flood models. Another option is to use a response-based approach, which involves simulating a large number of synthetic events from the multivariate statical model, run them through a flood model, and then calculate the empirical return water level at a given location using the water levels generated by these events. The latter is computationally demanding, possibly necessitating the use of a surrogate model, however the return level estimates are likely to be more robust than when adopting an event-based approach. In response to the Reviewer's comment, the last paragraph of Section 3.5 will be revised as follows:

"Although any combinations of NTR and RF along a given joint probability isoline have the same return probability, most hydrology-related engineering design approaches still rely on a single design event. Therefore, the "most likely event" strategy, introduced by Salvadori et al. (2011) and utilized in subsequent studies (e.g., Jane et al., 2020), is employed here. To quantify the relative probabilities of events along specific quantile isolines, we obtain $10^6$ combinations of NTR and RF by sampling from the fitted copulas, ensuring that the relative proportion of extremes is consistent with the empirical distribution. Then the relative probability along the isolines is calculated by the kernel density function of the simulated sample. The location of the "most likely" event is assigned to the point with the highest relative probability density on an isoline (Salvadori & Michele, 2013)."

The following part is also added to the discussion section that clarifies the reviewer's concern. (Where the limitations are discussed).

"However, analyzing only the most likely event (even though it may be the most plausible given the observations) does not capture the range of flood levels that could be generated by different combinations of flood drivers (NTR and RF) along an isoline. One way to address this limitation is to sample an ensemble of events (peak NTR-RF combinations) along the isoline and run them through flood models. Alternatively, a response-based approach can be employed, which involves simulating flood hazard for a large number of synthetic events from the multivariate statical model and then performing the statistical analysis on the response variable of interest (e.g., flood depth at a given location). The latter is computationally demanding, possibly necessitating the use of a surrogate model, however the return level estimates are likely to be more robust than when adopting an event-based approach (Jane et al., 2022). The simulated probabilistic flood depths and extents can then be incorporated with exposure and vulnerability data to perform a comprehensive flood risk assessment.

furthermore some minor comments in the attached document

line 74: Not sure I agree, see for example: https://link.springer.com/article/10.1007/s11069-024-06552-x

Thank you for pointing out the reference. It was published after we submitted our manuscript and hence couldn't include it. In our paper we claim that a "a comprehensive multivariate statistical framework for assessing compound flood potential" does not exist. The study that the Reviewer mentioned simulates the flood depths using physics-based models and applies univariate extreme value analysis to the simulated flood depths and combined yearly exceedance frequency of TCs, and ETCs following Dullaart et al. (2021). Therefore, this study does not really fall into the category of multivariate statistical frameworks assessing compound flood potential.

However, it is still relevant to the scope of our paper, and we now cite it in L46. An additional part was added to describe their work on L69 as follows:

"Nederhoff et al. (2024) addressed this aspect by employing a compound flood model for the coast of the United States, from Virginia to Florida. They separately simulated the total water levels induced by TCs and ETCs to assess their relative contributions and followed the approach outlined by Dullaart et al. (2021) to calculate the combined return water levels."

Line 141: Isn't this essentially the same as applying an areal reduction factor (ARF)?

Yes, the concept is similar to "areal reduction factor (ARF)". The ARF is generally applied to point rainfall estimates whenever an area (around the rain gauge) is large enough for rainfall not to be uniform. However, the Philadelphia International Airport rain gauge is not within the selected catchment (about 6 miles away from the selected catchment); thus, it is necessary to correct the bias due to the different locations (if any), additionally to what is corrected by ARF. Therefore, a simple method (quantile mapping) is used to transform rainfall gauge data to catchment representative quantities, and the term "bias-correction" is used to avoid any confusion.

Line 245: Why do you use sampling? You have the mathematical description of the copula and the marginals, so probability densities can be computed without sampling. Did you repeat this procedure multiple times with a different seed to test the variability in the results?

The authors agree that probability densities along the isoline can be calculated using the fitted copulas and marginals without sampling. However, this approach might not be straightforward since there are four copulas involved (conditioned on rainfall and NTR for both TC and non-TC) when estimating probability densities for the combined populations. Additionally, the same weight cannot be given to the relative joint return probabilities (along an isoline) derived from TC and non-TC copulas.

Therefore, a simpler method is employed here: sampling a sufficiently large number of NTR-rainfall combinations from each fitted copula to ensure that the relative proportion of extremes is consistent with the number of threshold exceedances in each corresponding sample. Then, the kernel density estimate is applied along the isoline for all the simulated NTR-rainfall combinations.

The sampling process was repeated multiple times, and it was noted that using 10,000 combinations is sufficient to obtain stable estimates for probability densities. However, following the Reviewer's comment and considering the low computation time for simulations, 1,000,000 simulations are now used to achieve even more stable estimates for probability densities along isolines. The text and Fig. 9 will be updated accordingly.

Line 313: To get more insight in the GOF for the highest values (extremes) a q-q plot could be valuable.

Thank you for the comment. The plots of parametric distributions with empirical distributions are used in the paper considering the importance of showing the confidence intervals. As per the Reviewers comment the quantile plots for each fitted distribution are shown below.

[Figure]

Figure R1: Q-Q plots for selected parametric distributions compared to the empirical distributions of Gloucester City, for TC events (left) and non-TC events (right). (a), (b) NTR POT events when conditioned on NTR. (c), (d) Maximum RF events corresponding to NTR POT events when conditioned on NTR. (e), (f) Maximum NTR events corresponding to RF POT events when conditioned on RF; (g), (h) RF POT events when conditioned on RF. The black line indicates the x=y line in each panel.

[Figure]

Figure R2: same as Figure R1, but for St. Petersburg

---

## Author Comment (AC3)

**Reviewer 3:**

I would like to compliment the authors for a well-written paper. The research gap this paper aims to fill is clear and relevant. Here are some minor comments to consider.

The authors would like to thank the Reviewer for their thoughtful comments. Below, we provide our responses to each comment and outline how we plan to adjust the manuscript following the NHESS review process.

- Consider adjusting the title to speak to a larger audience. Currently, the term 'mixed populations' does not speak for itself. Instead, it would be good to focus on the fact that you look at different storm types that can drive compound flooding.
    1. As a follow-up, the term 'populations' is never clearly defined.

Agree and the title will be changed as:

"A multivariate statistical framework for mixed storm types in compound flood analysis"

For the follow-up question: the sentence is revised from L54 to L55 as follows:

"However, most of these studies assume that all extreme events originate from a single population, which refers to a set of events or observations that share common characteristics and are generated by similar underlying processes"

- As RC1 and RC2 have highlighted there are some limitations and uncertainties in the methods that you have used. In the paper, I am missing a proper discussion of these limitations and how to overcome them in the discussion/conclusion.

Thank you for the comment. Two paragraphs are added to the end of the discussion, explaining the main limitations on L451 (of the original manuscript).

"One limitation of the proposed framework is the identification of compound events based on extreme flood drivers. In some locations, none of the flood drivers need to be extreme to cause compound flooding, as geographical exposure, and other factors (e.g., elevation, drainage, permeability) play dominant roles. Therefore, the focus here is on assessing the compound flood potential and how the joint probabilities of different flood drivers are linked to various storm types. To extend the proposed framework for fully characterizing the compound flood risk, the statistical approach can be combined with hydrodynamic numerical models (so called hybrid modeling, e.g., Moftakhari et al. 2019) to estimate flood inundation. However, analyzing only the most likely event (even though it may be the most plausible given the observations) does not capture the range of flood levels that could be generated by different combinations of flood drivers (NTR and RF) along an isoline. One way to address this limitation is to sample an ensemble of events (peak NTR-RF combinations) along the isoline and run them through flood models. Alternatively, a response-based approach can be employed, which involves simulating flood hazard for a large number of synthetic events from the multivariate statical model and then performing the statistical analysis on the response variable of interest (e.g., flood depth at a given location). The latter is computationally

demanding, possibly necessitating the use of a surrogate model, however the return level estimates are likely to be more robust than when adopting an event-based approach (Jane et al., 2022). The simulated probabilistic flood depths and extents can then be incorporated with exposure and vulnerability data to perform a comprehensive flood risk assessment.

Additionally, despite the long data records used (Gloucester City; 1901 to 2021 and St. Petersburg; 1948 to 2021), the stratified TC samples contain a relatively small number of events due to their rare occurrence in historical observations. When the framework is applied to a different site with less data, the smaller sample size may result in higher uncertainty in modeling the upper tail of the NTR and RF distributions. This limitation can be addressed by combining the proposed framework with synthetic flood driver information derived from physics-based models (e.g., Gori et al., 2020)."